# Interpreting Space-Based Trends in Carbon Monoxide with Multiple Models

Sarah A. Strode[1,2], Helen M. Worden[3], Megan Damon[2,4], Anne R. Douglass[2], Bryan N. Duncan[2], Louisa K. Emmons[3], Jean-Francois Lamarque[3], Michael Manyin[2,4], Luke D. Oman[2], Jose M. Rodriguez[2], Susan E. Strahan[1,2], Simone Tilmes[3]

[1]Universities Space Research Association, Columbia, MD, USA
[2]NASA Goddard Space Flight Center, Greenbelt, MD, USA
[3]National Center for Atmospheric Research, Boulder, CO, USA
[4]Science Systems and Applications, Inc., Lanham, MD, USA

*Correspondence to:* S. A. Strode (sarah.a.strode@nasa.gov)

## Abstract

We use a series of chemical transport model and chemistry climate model simulations to investigate the observed negative trends in MOPITT CO over several regions of the world, and to examine the consistency of time-dependent emission inventories with observations. We find that simulations driven by the MACCity inventory, used for the Chemistry Climate Modeling Initiative (CCMI), reproduce the negative trends in the CO column observed by MOPITT for 2000-2010 over the eastern United States and Europe. However, the simulations have positive trends over eastern China, in contrast to the negative trends observed by MOPITT. The model bias in CO, after applying MOPITT averaging kernels, contributes to the model-observation discrepancy in the trend over eastern China. This demonstrates that biases in a model's average concentrations can influence the interpretation of the temporal trend compared to satellite observations. The total ozone column plays a role in determining the simulated tropospheric CO trends. A large positive anomaly in the simulated total ozone column in 2010 leads to a negative anomaly in OH and hence a positive anomaly in CO, contributing to the positive trend in simulated CO. These results demonstrate that accurately simulating variability in the ozone column is important for simulating and interpreting trends in CO.

## 1. Introduction

Carbon monoxide (CO) is an air pollutant that contributes to ozone formation and
affects the oxidizing capacity of the troposphere (Thompson, 1992; Crutzen, 1973).  Its
primary loss is through reaction with OH, which leads to a lifetime of 1-2 months (Bey et
al., 2001) and makes CO an excellent tracer of long-range transport.  Both fossil fuel
combustion and biomass burning are major sources of CO.  The biomass burning source
shows large interannual variability (van der Werf et al., 2010), while fossil fuel emissions
typically change more gradually.  The time-dependent MACCity inventory (Granier et
al., 2011) shows decreases in CO emissions from the United States and Europe from
2000 to 2010 due to increasing pollution controls, but increases in emissions from China.
MACCity emissions for years after 2000 are based on the Representative Concentration
Pathway (RCP) 8.5 (Riahi et al., 2007).  The REAS (Kurokawa et al., 2013) and
EDGAR4.2 (EC-JRC/PBL, 2011) inventories also show increasing CO emissions from
China.  The bottom-up inventory of Zhang et al. (2009) shows an 18% increase in CO
emissions from China from 2001 to 2006, and Zhao et al. (2012) estimate a 6% increase
between 2005 and 2009.  However, there is considerable uncertainty in bottom-up
inventories, and comparison of model hindcast simulations driven by bottom-up
inventories with observations provides an important test of the time-dependent emission
estimates.

Space-based observations of CO are now available for over a decade and show
trends at both hemispheric and regional scales.  Warner et al. (2013) found significant
negative trends in both background CO and recently emitted CO at 500 hPa over southern
hemisphere oceans and northern hemisphere land and ocean in Atmospheric Infrared
Sounder (AIRS) data.  Worden et al. (2013) calculated trends in the CO column from
several thermal infrared (TIR) instruments including MOPITT and AIRS.  They found
statistically significant negative trends over Europe, the eastern United States, and China
for 2002-2012.  He et al. (2013) also report a negative trend in MOPITT near-surface CO
over western Maryland.

Surface concentrations of CO show downward trends over the United States
driven by emission reductions (EPA, 2011), consistent with the space-based trends.
Decreases in the partial column of CO from FTIR stations in Europe also show decreases
from 1996 to 2006, consistent with emissions decreases (Angelbratt et al., 2011).  Yoon
and Pozzer (2014) found that a model simulation of 2001 to 2010 reproduced negative
trends in surface CO over the eastern U.S. and western Europe, but showed a positive
trend in surface CO over southern Asia.

The cause of the negative trend over China seen in MOPITT and AIRS data is

uncertain.  The trend is consistent with the results of Li and Liu (2011), who found
decreases in surface CO measurements in Beijing, and with decreases in CO emissions in
2008 inferred from the correlation of CO with $CO_2$ measured at Hateruma Island
(Tohjima et al., 2014) and at a rural site in China (Wang et al., 2010).  Yumimoto et al.
(2014) used inverse modeling of MOPITT data to infer a decrease in CO emissions from
China after 2007.  The 2008 Olympic Games and the 2009 global economic slowdown
led to reductions in CO (Li and Liu, 2011; Worden et al., 2012).  However, the negative
trend in MOPITT CO is inconsistent with the rising CO emissions of the MACCity and
REAS inventories.   Inverse modeling of MOPITT version 6 data yields a negative trend
in CO emissions from China and a larger global decline in CO emissions than that found
in the MACCity inventory (Yin et al., 2015).

This study examines whether global hindcast simulations can reproduce the trends

and variability in carbon monoxide seen in the MOPITT record.   We examine the role of
averaging kernels and the contribution of trends at different altitudes to the trends
observed by MOPITT.  We then examine the impact of OH variability on the simulated
trends in CO.

## 84  2. Methods

### 85  2.1. MOPITT

The MOPITT instrument onboard the Terra Satellite provides the longest satellite-

based record of atmospheric CO, with observations available from March 2000 to
present.  It provides nearly global coverage every three days (Edwards et al., 2004).  We
use the monthly Level 3 daytime column data from the Version 5 TIR product, which has
negligible drift in the bias over time (Deeter et al., 2013).  The level 3 data is a gridded
product and includes the a priori and averaging kernel for each grid box.  Supplemental
Figure S1 shows the MOPITT column averaging kernels averaged over four regions.  The
column averaging kernels depend on the observed scene, and vary year to year as well as
seasonally.  The dependence of the column averaging kernels on the CO mixing ratio
profile (Deeter, 2009) explains the high values in the lower troposphere over eastern
China in winter.

We calculate trends and de-seasonalized anomalies for the Eastern U.S., Europe, and

eastern China regions described by Worden et al. (2013).  Trends that differ from zero by
more than the two-sigma uncertainty on the trend are considered statistically significant.
We account for autocorrelation of the data for a one-month lag when calculating the
uncertainty on the trends.  We calculate the annual cycle by fitting the data with a series
of sines and cosines as well as the linear trend, and then remove the annual cycle to
obtain the de-seasonalized anomalies.  Months with no MOPITT data or only a few days
of MOPITT data are excluded from the trend analysis.  This includes May-August of
2001 and August-September of 2009.  We report the MOPITT trends for 2000-2010 for
comparison with model simulations, and for 2000-2014 to give a longer-term view of the
observed trends.

**2.2.  Model Simulations**

We use a suite of chemistry climate model (CCM) and chemical transport model

(CTM) simulations to interpret the observed trends.  The Global Modeling Initiative
(GMI) CTM includes both tropospheric (Duncan et al., 2007) and stratospheric (Strahan
et al., 2007) chemistry, including over 400 reactions and 124 chemical species.
Meteorology for the GMI simulations comes from the Modern-Era Retrospective
Analysis for Research and Applications (MERRA) (Rienecker et al., 2011).  The GEOS-
5 Chemistry Climate Model (GEOSCCM)(Oman et al., 2011) incorporates the GMI
chemical mechanism into the GEOS-5 atmospheric general circulation model (AGCM).
The GEOSCCM simulations are forced by observed sea surface temperatures (SSTs)
from (Reynolds et al., 2002).

The Community Earth System Model, CESM1 CAM4-chem, includes 191 chemical

tracers and over 400 reactions for both troposphere and stratosphere (Tilmes et al., 2016).
The model can be run fully coupled to a free-running ocean, with prescribed SSTs, or
with nudged meteorology from GEOS-5 or MERRA analysis. CESM1 CAM4-chem is
further coupled to the land model, providing biogenic emissions from the Model of
Emissions and Aerosols from Nature (MEGAN), version 2.1 (Guenther et al., 2012).
Several simulations were conducted as part of the Chemistry-Climate Model Initiative
(CCMI) project (Eyring et al., 2013). These include the Ref-C1 simulation of the
GEOSCCM and a Ref-C1 CESM1 CAM4-Chem simulation, hereafter called G-Ref-C1
and C-Ref-C1, respectively, and the Ref-C1-SD simulation of the GMI CTM. Both the
Ref-C1 and the Ref-C1-SD simulations use time-dependent anthropogenic and biomass
burning emissions from the MACCity inventory (Granier et al., 2011), but the Ref-C1-
SD simulations use specified meteorology while the Ref-C1 simulations run with
prescribed SSTs. The MACCity inventory linearly interpolates the decadal
anthropogenic emissions from the ACCMIP inventory (Lamarque et al., 2010) for 2000,
and the RCP8.5 emissions for 2005 and 2010, to each year in between. The MACCity
biomass burning emissions have year-to-year variability based on the GFED-v2 (van der
Werf et al., 2006) inventory. From 2000 to 2010, CO emissions in the MACCity
inventory decreased from 31 to 11 Tg yr$^{-1}$ over the eastern U.S., from 97 to 59 Tg yr$^{-1}$
over Europe, and increased from 56 Tg to 72 Tg yr$^{-1}$ over eastern China.
Given the uncertainty in CO emissions, we conduct a GMI CTM simulation using an
alternative time-dependent emissions scenario, called AltEmis. This simulation is
described in detail in (Strode et al., 2015b). Briefly, anthropogenic emissions include
time-dependence based on EPA (http://www.epa.gov/ttn/chief/trends/index.html), the
REAS inventory (Ohara et al., 2007), and EMEP
(http://www.ceip.at/ms/ceip_home1/ceip_home/webdab_emepdatabase/reported_emissio
ndata/), and annual scalings from van Donkelaar et al. (2008). Biomass burning
emissions are based on the GFED3 inventory (van der Werf et al., 2010). While the
regional emission trends in this simulation are of the same sign as in the Ref-C1 case, the
magnitude of the negative trends over the U.S. and Europe are smaller and the positive
trend over China is larger, leading to a positive global trend (Fig. 1). We also conduct a
sensitivity study called EmFix with anthropogenic and biomass burning emissions held
constant at year 2000 levels. Table 1 summarizes the simulations used in this study.
We regrid the model output to the MOPITT grid and convolve the simulated CO with
the MOPITT averaging kernels and a priori in order to compare the simulated and
observed CO columns. The averaging kernels are space and time dependent. We use the
following equation from Deeter et al. (2013):

$C_{sim} = C_0 + \mathbf{a}(\mathbf{x}_{mod} - \mathbf{x}_0)$       (1)

where $C_{sim}$ and $C_0$ are the simulated and a priori CO total columns, respectively, $\mathbf{a}$ is the
total column averaging kernel, and $\mathbf{x}_{mod}$ and $\mathbf{x}_0$ are the modeled and a priori CO profiles,
respectively. The column averaging kernel is calculated from the standard averaging
kernel matrix, which is based on the log of the CO concentration profile, following the
method of Deeter (2009):

$a_j = (K / \log_{10}e) \sum \Delta p_i \, v_{rtv,i} \, A_{ij}$       (2)

where $\Delta p_i$ and $v_{rtv,i}$ are the pressure thickness and retrieved CO concentration,
respectively, of level i, $\mathbf{A}$ is the standard averaging kernel matrix, and $K = 2.12 * 10^{13}$
molec $cm^{-2}$ $hPa^{-1}$ $ppb^{-1}$.

We deseasonalize the simulated CO columns and calculate their linear trend

following the same procedure that we applied to the MOPITT CO. Months that do not
have MOPITT data (June-July 2001 and August-September 2009) are excluded from the
analysis of the model trends as well.
The Ref-C1 and Ref-C1-SD simulations requested by CCMI extend until 2010.
However, the MACCity biomass burning emissions extend only until 2008. CAM4-
Chem therefore repeated the biomass burning emissions for 2008 for years 2009-2010.
In contrast, the GEOSCCM Ref-C1 and GMI Ref-C1-SD simulations used emissions
from GFED3 (van der Werf et al., 2010) for years after 2008. Some simulations were
available through 2011, while others ended in 2010. We therefore report results for
2000-2010, but note that extending the analysis through 2011 does not alter the
conclusions.
**3. Results**
**3.1. Trends over Europe, the United States, and the Northern Hemisphere**
The hindcast simulations driven by MACCity emissions (G-Ref-C1, Ref-C1-SD, and
C-Ref-C1) show negative trends in CO over the U.S. and Europe that agree with the
observed slope from MOPITT within the uncertainty (Fig. 2, Table 2). The MOPITT
trends for both regions are statistically significant for both regions, as shown by Worden
et al. (2013).  These results are consistent with the findings of Yin et al. (2015), whose
inversion of MOPITT data showed a posteriori trends in CO emissions over the U.S. and
western Europe that were consistent with but slightly larger than the a priori trends.  The
EmFix hindcast shows a positive, though non-significant, trend for both regions,
indicating that the decrease in CO emissions is necessary for reproducing the downward
trend in the CO column.  The AltEmis simulation fails to produce the negative trends,
despite including negative trends in regional emissions for both the U.S. and Europe.
The impact of these negative regional trends is insufficient to overcome the positive
global emission trend in the AltEmis scenario (Fig. 1), leading to positive trends in CO.
Figure 2 also reveals a negative bias in the simulated CO column between the models
and MOPITT.  A low bias in simulated CO at northern latitudes is often present in global
models (Naik et al., 2013), and may indicate a high bias in northern hemisphere OH
(Strode et al., 2015a) or CO dry deposition (Stein et al., 2014), as well as an
underestimate of CO emissions.
The deseasonalized anomalies in the MOPITT and simulated CO columns are shown
in Fig. 2b,d, and the correlation coefficient between the observed and simulated monthly
anomalies are presented in Table 2b.  The highest correlations are for the AltEmis and
Ref-C1-SD simulations of the GMI CTM.  This result is consistent with the use of year-
specific meteorology, which we expect to better match the transport of particular years.
The lowest correlations are for the EmFix simulation.  This is expected since the EmFix
simulation does not include inter-annual variability (IAV) in biomass burning.  The IAV
in biomass burning makes a large contribution to the IAV of CO (Voulgarakis et al.,

2015).

The role of biomass burning in driving the CO variability is even more evident at the
hemispheric scale.  Figure 2g,h shows the anomalies in MOPITT and the simulations for
the northern hemisphere (0-60N). The EmFix simulation shows almost no correlation,
while the other simulations have correlation coefficients exceeding 0.6 (Table 2).   The
role of changing anthropogenic emissions is also evident, as the Ref-C1-SD simulation
captures the 2008-2009 dip in the CO column while the EmFix simulation does not.
Gratz et al. (2015) found decreasing CO concentrations at Mount Bachelor Observatory
in Oregon during spring for 2004-2013, which they attribute to reductions in emissions
leading to a lower hemispheric background.   We also note that Ref-C1-SD and G-Ref-
C1 have similar correlations with the observed variability for the northern hemisphere
(Table 2), indicating that transport differences are less important for variability at the
hemispheric scale.
**3.2.  Trend over China**

Observations from MOPITT show a negative trend in the CO column over eastern

China for 2002-2012 (Worden et al., 2013).  The negative trend for the years 2000-2014
exceeds that for 2000-2010 (Table 2), showing that it is not driven solely by temporary
emission reductions in 2008.  Our simulations do not reproduce this trend, and instead
show increases in the CO column (Fig. 2e), which is expected given that CO emissions
from China increase in four of the five simulations.  The anomalies (Fig. 2f) show that
the discrepancy in the simulated versus observed trends is driven largely by the failure of
the simulations to capture the 2008 dip in the CO column, leading to an overestimate that
continues through 2010.  This suggests emission reductions in China during this time
period are not adequately captured by the emission inventories.  However, the good
agreement between the observed and simulated decreases in CO for the northern
hemisphere as a whole (Fig. 2g,h) suggest that on a global scale, the emission time series
is reasonable.  Consequently, we examine several other factors that may contribute to the
difference in sign between the MOPITT and simulated CO trends.

Regional trends in CO are expected to vary with altitude, with surface concentrations

most heavily influenced by local emissions.  MOPITT TIR retrievals have higher
sensitivity to CO in the mid-troposphere than at the surface (Deeter et al., 2004), so the
trend in the MOPITT CO column will be weighted towards the trends in free tropospheric
CO rather than near-surface CO.  We quantify this impact on our Ref-C1-SD CO column
trends by comparing the trend in the pure-model CO column with that of the simulated
column convolved with the MOPITT averaging kernels.

The simulated CO trend over eastern China for 2000-2010 is positive (but not

significant) both with and without the averaging kernels, but application of the MOPITT
kernels increases the positive trend from $1.3*10^{16}$ molec cm$^{-2}$ yr$^{-1}$ to $1.4*10^{16}$ molec cm$^{-2}$
yr$^{-1}$.  This result is initially surprising since we expect trends in the mid-troposphere to be
more strongly influenced by the decrease in the hemispheric CO background. Indeed, the
trends in CO concentration over eastern China simulated in Ref-C1-SD switch from
positive in the lower troposphere to negative in the middle and upper troposphere.
However, the application of the kernels results in more positive (or less negative) trends
in all regions.

Yoon et al. (2013) show that since the averaging kernels vary over time, a bias

between the true atmosphere and the a priori assumed by MOPITT can lead to an
artificial trend in the retrieved CO. Similarly, the bias between the average simulated CO
concentrations and the MOPITT a priori, evident in Figure 2, can lead to an artifact in the
simulated CO trend when the simulation is convolved with the MOPITT averaging
kernels. This is due to the changing contribution of the a priori when the vertical
sensitivity (averaging kernel) is varying in time. MOPITT vertical sensitivity varies with
time due to instrument degradation as well as the change in CO abundance. The bias in
CO varies with altitude, so if the vertical sensitivity described by the averaging kernel
changes, this will change the value of the convolved CO column even if there were no
changes in the CO profile. Furthermore, changes in the averaging kernel result in more
or less weight placed on the a priori versus the CO simulated by the model. Thus, a
difference between the a priori and the model means that placing more (or less) weight on
the a priori will change the resulting value of $C_{sim}$. Since the a priori profiles and
columns are constant in time, taking the time derivative of equation 1 yields:

$\partial C_{sim}/\partial t = \mathbf{a}\,(\partial \mathbf{x}_{mod}/\partial t) + \partial \mathbf{a}/\partial t\,(\mathbf{x}_{mod} - \mathbf{x}_0)$      (3)

The second term on the right hand side shows that the larger the bias between the
modeled CO and the a priori, the larger the impact of the changing averaging kernel.

We quantify this effect by convolving the simulated CO for each year with the

MOPITT averaging kernels for the year 2008, thus removing the effect of the time-
dependence of the averaging kernels. The resulting trend, $0.56*10^{16}$ molec cm$^{-2}$ yr$^{-1}$, is
less positive than the pure model trend or the original simulated trend. Thus, accounting
for the time-dependence of the averaging kernels convolved with model bias reduces but
does not eliminate the discrepancy with the observed trend. Comparing the trend for the
constant averaging kernel case with the original simulated trend for Ref-C1-SD $(1.4*10^{16}$
molec cm$^{-2}$ yr$^{-1})$ suggests that the changing averaging kernels combined with the model
bias contribute $0.84*10^{16}$ molec cm$^{-2}$ yr$^{-1}$ to the simulated trend. Other regions also show
a more negative trend when the same averaging kernel is applied to the model results for
all years. The large bias in CO at middle and high northern latitudes commonly seen in
modeling studies thus impacts the ability of models to reproduce and attribute observed
trends in satellite data.
Figure 2 and Table 2 also show a positive trend in the GMI EmFix simulation for
eastern China. This larger trend in the EmFix simulation than the Ref-C1-SD simulation
indicates that the net decrease in emissions contributes to decreasing CO over eastern
China, consistent with the observed negative trend, but other factors in the model cause
an increase in CO over eastern China even when all emissions are constant. Subtracting
the EmFix trend from the Ref-C1-SD trend shows that the changing emissions contribute
a CO trend of -0.7 molec cm$^2$ yr$^{-1}$ over eastern China. The 2.1 molec cm$^2$ yr$^{-1}$ trend in the
EmFix simulation, which reflects the impacts of the simulated chemistry and transport,
thus contributes to the erroneous sign of the trend in the GMI simulations. The trends in
the EmFix simulation for the northern hemisphere average and the eastern U.S. and
Europe are positive as well (Table 2). We examine their cause in the next section.
**3.3. Contribution of OH Interannual Variability**

Since the EmFix simulation shows a positive trend in the northern hemisphere, we
next examine the variability in the CO sink, OH. We also examine variability in the total
ozone column, since overhead ozone is a major driver of OH variability (Duncan and
Logan, 2008). Figure 3 shows the variability in CO and OH in the EmFix simulation.
The positive and negative anomalies in CO correspond with the negative and positive
anomalies, respectively, in OH. The anomalies in OH are in turn inversely related to
anomalies in the total ozone column. The correlation coefficient between OH and
column ozone is -0.53 for the 15°S-15°N average, -0.72 for the 15°-25°N average, and -
0.75 for the 30°-60°N average. The large NH ozone anomaly in 2010, in particular, leads
to a large anomaly in OH and thus CO. This OH anomaly extends from the northern
tropics to the midlatitudes. The large CO anomaly near the end of the time series
contributes to the apparent 11-year trend. We note that since the lifetime of CO is several
months, CO anomalies are not expected to have a one-to-one correspondence with the
OH anomalies.

The large anomaly in the simulated total ozone column in 2010 is overestimated

compared to observations.  Figure 4 shows the time-dependence of the total ozone
column from 30°-60°N in EmFix compared to SBUV data (Frith et al., 2014).  While the
observations show an anomaly in 2010, the magnitude is smaller than that produced by
the simulation.  Steinbrecht et al. (2011) attribute the 2010 anomaly in northern
midlatitude ozone observations to a combination of an unusually strong negative Arctic
Oscillation and North Atlantic Oscillation and the easterly phase of the quasi-biennial
oscillation.

While the impact of OH interannual variability on the apparent trend in CO is clear in

the EmFix simulation, this source of variability is partially masked by large interannual
variability in CO emissions in the other simulations.  We examine the correlation
between the de-trended and deseasonalized CO anomalies from 10°S-10°N in the Ref-
C1-SD simulation and the CO emissions as well as the simulated OH and column ozone.
Since the CO emitted in a given month can influence concentrations for several
subsequent months, we use a 3-month smoothing of the emission time series.  We find a
high correlation (r=0.88) between the CO anomalies and the CO emissions.  This
correlation is also evident in the MOPITT data, as the MOPITT CO anomalies have a
correlation of r=0.70 with the emissions.  Figure 5 shows the strong relationship between
the simulated CO anomalies and the CO emissions.  However, the colors in Fig. 5
indicate that the scatter for a given level of emissions is often linked to the OH
anomalies, with low/high OH anomalies leading to CO that is higher/lower than would be
predicted just from the CO emissions.  We find that the 10°S-10°N OH in the Ref-C1-SD
simulation is anticorrelated with CO (r=-0.62) and with the total ozone column (r=-0.68).
Consequently, the simulated ozone column plays a role in modulating tropical CO
variability even when variable CO emissions are included, although the emissions still
play the strongest role.

## 4. Conclusions

We conducted a series of multi-year simulations to analyze the causes of the negative trends in MOPITT CO reported by Worden et al. (2013). Both CTM and CCM simulations driven by the MACCity emissions reproduce the observed trends over the eastern U.S. and Europe, providing confidence in the regional emission trends.

None of the simulations reproduce the observed negative trend over eastern China. This negative trend persists even with the MOPITT data extended out to 2014. The MOPITT averaging kernels are weighted towards the free troposphere, where the relative importance of hemispheric versus local trends is greater. However, our simulations indicate that this effect is insufficient to explain the negative trends over China. Indeed, the negative trend in MOPITT CO over eastern China ($-2.9*10^{16}$ molec cm$^{-2}$ yr$^{-1}$) is stronger than that of the northern hemisphere average ($-1.4*10^{16}$ molec cm$^{-2}$ yr$^{-1}$), indicating that changes in hemispheric CO account for less than half of the trend over China. While the simulations' underestimate of the observed trend likely indicates a too positive emission trend for China, several other factors play a role in the model-observation mismatch. We find that the time-dependent MOPITT averaging kernels, combined with the low bias in simulated CO, provides a positive component to the simulated trends. Large anomalies in the simulated ozone column in the GMI CTM simulations also contribute a positive component to the northern hemisphere trends due to their impact on OH. For the Ref-C1-SD simulation, the trends due to the model bias combined with changing averaging kernels ($0.84*10^{16}$ molec cm$^{-2}$ yr$^{-1}$) and to the simulated chemistry and transport ($2.1*10^{16}$ molec cm$^{-2}$ yr$^{-1}$) can together account for almost 70% of the $4.3*10^{16}$ molec cm$^{-2}$ yr$^{-1}$ difference between the Ref-C1-SD and MOPITT trends over eastern China.

Variability in emissions is the primary driver of year-to-year variability in simulated CO, but OH variability also plays a role. The simulated OH is anti-correlated with both CO and the total ozone column, highlighting the importance of realistic overhead ozone columns for accurately simulating CO variability and trends. In addition, further work is needed to understand recent changes in CO emissions from China.

**Acknowledgements**

This work was supported by NASA's Modeling, Analysis, and Prediction program
and computing resources from the NASA High-End Computing Program.  We thank
Bruce Van Aartsen for contributing to the GMI simulations.  The CESM project is
supported by the National Science Foundation and the Office of Science (BER) of the US
Department of Energy. The MOPITT project is supported by the NASA Earth Observing
System (EOS) Program. The National Center for Atmospheric Research (NCAR) is
sponsored by the National Science Foundation.

Angelbratt, J., Mellqvist, J., Simpson, D., Jonson, J., Blumenstock, T., Borsdorff, T.,
Duchatelet, P., Forster, F., Hase, F., Mahieu, E., De Maziere, M., Notholt, J., Petersen,
A., Raffalski, U., Servais, C., Sussmann, R., Warneke, T., and Vigouroux, C.: Carbon
monoxide (CO) and ethane (C2H6) trends from ground-based solar FTIR measurements
at six European stations, comparison and sensitivity analysis with the EMEP model,
Atmospheric Chemistry and Physics, 11, 9253-9269, 10.5194/acp-11-9253-2011, 2011.
Bey, I., Jacob, D., Logan, J., and Yantosca, R.: Asian chemical outflow to the Pacific in
spring: Origins, pathways, and budgets, Journal of Geophysical Research-Atmospheres,
106, 23097-23113, 10.1029/2001JD000806, 2001.
Crutzen, P.: A Discussion of the Chemistry of Some Minor Constituents in the
Stratosphere and Troposphere, Pure and Applied Geophysics, 106, 1385-1399,
10.1007/BF00881092, 1973.
Deeter, M., Emmons, L., Edwards, D., Gille, J., and Drummond, J.: Vertical resolution
and information content of CO profiles retrieved by MOPITT, Geophysical Research
Letters, 31, 10.1029/2004GL020235, 2004.
Deeter, M. N.: MOPITT (Measurements of Pollution in the Troposphere) Validated
Version 4 Product User's Guide, National Center for Atmospheric Research.  Available
at http://web3.acd.ucar.edu/mopitt/v4_users_guide_val.pdf, 2009.
Deeter, M. N., Martinez-Alonso, S., Edwards, D. P., Emmons, L. K., Gille, J. C.,
Worden, H. M., Pittman, J. V., Daube, B. C., and Wofsy, S. C.: Validation of MOPITT
Version 5 thermal-infrared, near-infrared, and multispectral carbon monoxide profile
retrievals for 2000-2011, Journal of Geophysical Research-Atmospheres, 118, 6710-
6725, 10.1002/jgrd.50272, 2013.
Duncan, B. N., Strahan, S. E., Yoshida, Y., Steenrod, S. D., and Livesey, N.: Model study
of the cross-tropopause transport of biomass burning pollution, Atmospheric Chemistry
and Physics, 7, 3713-3736, 2007.
Duncan, B. N., and Logan, J. A.: Model analysis of the factors regulating the trends and
variability of carbon monoxide between 1988 and 1997, Atmospheric Chemistry and
Physics, 8, 7389-7403, 2008.
Edwards, D. P., Emmons, L. K., Hauglustaine, D. A., Chu, D. A., Gille, J. C., Kaufman,
Y. J., Petron, G., Yurganov, L. N., Giglio, L., Deeter, M. N., Yudin, V., Ziskin, D. C.,
Warner, J., Lamarque, J. F., Francis, G. L., Ho, S. P., Mao, D., Chen, J., Grechko, E. I.,
and Drummond, J. R.: Observations of carbon monoxide and aerosols from the Terra
satellite: Northern Hemisphere variability, Journal of Geophysical Research-
Atmospheres, 109, 10.1029/2004jd004727, 2004.
EPA: Our Nation's Air - Status and Trends through 2010, edited by: EPA-454/R-12-001,
Research Triangle Park, NC, 2011.
Eyring, V., Lamarque, J.-F., Hess, P., Arfeuille, F., Bowman, K., Chipperfiel, M. P.,
Duncan, B., Fiore, A., Gettelman, A., and Giorgetta, M. A.: Overview of IGAC/SPARC
Chemistry-Climate Model Initiative (CCMI) community simulations in support of
upcoming ozone and climate assessments, Sparc Newsletter, 40, 48-66, 2013.
Frith, S., Kramarova, N., Stolarski, R., McPeters, R., Bhartia, P., and Labow, G.: Recent
changes in total column ozone based on the SBUV Version 8.6 Merged Ozone Data Set,
Journal of Geophysical Research: Atmospheres, 119, 9735-9751, 2014.
Granier, C., Bessagnet, B., Bond, T., D'Angiola, A., van der Gon, H. D., Frost, G. J.,
Heil, A., Kaiser, J. W., Kinne, S., Klimont, Z., Kloster, S., Lamarque, J. F., Liousse, C.,
Masui, T., Meleux, F., Mieville, A., Ohara, T., Raut, J. C., Riahi, K., Schultz, M. G.,
Smith, S. J., Thompson, A., van Aardenne, J., van der Werf, G. R., and van Vuuren, D.
P.: Evolution of anthropogenic and biomass burning emissions of air pollutants at global
and regional scales during the 1980-2010 period, Climatic Change, 109, 163-190,
10.1007/s10584-011-0154-1, 2011.
Gratz, L., Jaffe, D., and Hee, J.: Causes of increasing ozone and decreasing carbon
monoxide in springtime at the Mt. Bachelor Observatory from 2004 to 2013,
Atmospheric Environment, 109, 323-330, 10.1016/j.atmosenv.2014.05.076, 2015.
Guenther, A., Jiang, X., Heald, C., Sakulyanontvittaya, T., Duhl, T., Emmons, L., and
Wang, X.: The Model of Emissions of Gases and Aerosols from Nature version 2.1
(MEGAN2. 1): an extended and updated framework for modeling biogenic emissions,
432  2012.
He, H., Stehr, J., Hains, J., Krask, D., Doddridge, B., Vinnikov, K., Canty, T., Hosley, K.,
Salawitch, R., and Worden, H.: Trends in emissions and concentrations of air pollutants
in the lower troposphere in the Baltimore/Washington airshed from 1997 to 2011,
Atmospheric Chemistry and Physics, 13, 7859-7874, 2013.
Kurokawa, J., Ohara, T., Morikawa, T., Hanayama, S., Janssens-Maenhout, G., Fukui, T.,
Kawashima, K., and Akimoto, H.: Emissions of air pollutants and greenhouse gases over
Asian regions during 2000-2008: Regional Emission inventory in ASia (REAS) version
2, Atmospheric Chemistry and Physics, 13, 11019-11058, 10.5194/acp-13-11019-2013,
441  2013.
Lamarque, J.F., Bond, T.C., Eyring, V., Granier, C., Heil, A., Klimont, Z., Lee, D.,
Liousse, C., Mieville, A., Owen, B. and Schultz, M.G.: Historical (1850–2000) gridded
anthropogenic and biomass burning emissions of reactive gases and aerosols:
methodology and application, Atmospheric Chemistry and Physics, 10, 7017-7039, 2010.
Li, L., and Liu, Y.: Space-borne and ground observations of the characteristics of CO
pollution in Beijing, 2000–2010, Atmospheric Environment, 45, 2367-2372,
http://dx.doi.org/10.1016/j.atmosenv.2011.02.026, 2011.
Naik, V., Voulgarakis, A., Fiore, A. M., Horowitz, L. W., Lamarque, J. F., Lin, M.,
Prather, M. J., Young, P. J., Bergmann, D., Cameron-Smith, P. J., Cionni, I., Collins, W.
J., Dalsoren, S. B., Doherty, R., Eyring, V., Faluvegi, G., Folberth, G. A., Josse, B., Lee,
Y. H., MacKenzie, I. A., Nagashima, T., van Noije, T. P. C., Plummer, D. A., Righi, M.,
Rumbold, S. T., Skeie, R., Shindell, D. T., Stevenson, D. S., Strode, S., Sudo, K., Szopa,

454 S., and Zeng, G.: Preindustrial to present-day changes in tropospheric hydroxyl radical
455 and methane lifetime from the Atmospheric Chemistry and Climate Model
456 Intercomparison Project (ACCMIP), Atmospheric Chemistry and Physics, 13, 5277-
457 5298, 10.5194/acp-13-5277-2013, 2013.
458 Ohara, T., Akimoto, H., Kurokawa, J., Horii, N., Yamaji, K., Yan, X., and Hayasaka, T.:
459 An Asian emission inventory of anthropogenic emission sources for the period 1980-
460 2020, Atmospheric Chemistry and Physics, 7, 4419-4444, 2007.
461 Oman, L. D., Ziemke, J. R., Douglass, A. R., Waugh, D. W., Lang, C., Rodriguez, J. M.,
462 and Nielsen, J. E.: The response of tropical tropospheric ozone to ENSO, Geophysical
463 Research Letters, 38, 10.1029/2011gl047865, 2011.
464 Reynolds, R., Rayner, N., Smith, T., Stokes, D., and Wang, W.: An improved in situ and
465 satellite SST analysis for climate, Journal of Climate, 15, 1609-1625, 10.1175/1520-
466 0442(2002)015<1609:AIISAS>2.0.CO;2, 2002.
467 Riahi, K., Grübler, A., and Nakicenovic, N.: Scenarios of long-term socio-economic and
468 environmental development under climate stabilization, Technological Forecasting and
469 Social Change, 74, 887-935, 2007.
470 Rienecker, M. M., Suarez, M. J., Gelaro, R., Todling, R., Bacmeister, J., Liu, E.,
471 Bosilovich, M. G., Schubert, S. D., Takacs, L., Kim, G.-K., Bloom, S., Chen, J., Collins,
472 D., Conaty, A., da Silva, A., Gu, W., Joiner, J., Koster, R. D., Lucchesi, R., Molod, A.,
473 Owens, T., Pawson, S., Pegion, P., Redder, C. R., Reichle, R., Robertson, F. R., Ruddick,
474 A. G., Sienkiewicz, M., and Woollen, J.: MERRA: NASA's Modern-Era Retrospective
475 Analysis for Research and Applications, Journal of Climate, 24, 3624-3648,
476 10.1175/JCLI-D-11-00015.1, 2011.
477 Stein, O., Schultz, M., Bouarar, I., Clark, H., Huijnen, V., Gaudel, A., George, M., and
478 Clerbaux, C.: On the wintertime low bias of Northern Hemisphere carbon monoxide
479 found in global model simulations, Atmospheric Chemistry and Physics, 14, 9295-9316,
480 10.5194/acp-14-9295-2014, 2014.
481 Steinbrecht, W., Köhler, U., Claude, H., Weber, M., Burrows, J.P., and van der A, R.J.:
482 Very high ozone columns at northern mid-latitudes in 2010, Geophysical Research
483 Letters, 38, 10.1029/2010GL046634, 2011.
484 Strahan, S. E., Duncan, B. N., and Hoor, P.: Observationally derived transport diagnostics
485 for the lowermost stratosphere and their application to the GMI chemistry and transport
486 model, Atmospheric Chemistry and Physics, 7, 2435-2445, 2007.
487 Strode, S., Duncan, B., Yegorova, E., Kouatchou, J., Ziemke, J., and Douglass, A.:
488 Implications of carbon monoxide bias for methane lifetime and atmospheric composition
489 in chemistry climate models, Atmospheric Chemistry and Physics, 15, 11789-11805,
490 2015a.
491 Strode, S. A., Rodriguez, J. M., Logan, J. A., Cooper, O. R., Witte, J. C., Lamsal, L. N.,
492 Damon, M., Van Aartsen, B., Steenrod, S. D., and Strahan, S. E.: Trends and variability
493 in surface ozone over the United States, Journal of Geophysical Research: Atmospheres,
494 120, 9020-9042, 2015b.
495 Thompson, A.: The Oxidizing Capacity of the Earth's Atmosphere: Probable Past and
496 Future Change, Science, 256, 1157-1165, 10.1126/science.256.5060.1157, 1992.
497 Tilmes, S., Lamarque, J. F., Emmons, L. K., Kinnison, D. E., Marsh, D., Garcia, R. R.,
498 Smith, A. K., Neely, R. R., Conley, A., Vitt, F., Val Martin, M., Tanimoto, H., Simpson,
499 I., Blake, D. R., and Blake, N.: Representation of the Community Earth System Model

(CESM1) CAM4-chem within the Chemistry-ClimateModel Initiative (CCMI), Geosci. Model Dev. Discuss., 2016, 1-50, 10.5194/gmd-2015-237, 2016.

Tohjima, Y., Kubo, M., Minejima, C., Mukai, H., Tanimoto, H., Ganshin, A., Maksyutov, S., Katsumata, K., Machida, T., and Kita, K.: Temporal changes in the emissions of $CH_4$ and CO from China estimated from $CH_4/CO_2$ and $CO/CO_2$ correlations observed at Hateruma Island, Atmospheric Chemistry and Physics, 14, 1663-1677, 10.5194/acp-14-1663-2014, 2014.

van der Werf, G. R., Randerson, J. T., Giglio, L., Collatz, G. J., Kasibhatla, P.S., and Arellano, Jr., A.F.: Interannual variability in global biomass burning emissions from 1997-2004, Atmospheric Chemistry and Physics, 6, 3423-3441, 2006.

van der Werf, G. R., Randerson, J. T., Giglio, L., Collatz, G. J., Mu, M., Kasibhatla, P. S., Morton, D. C., DeFries, R. S., Jin, Y., and van Leeuwen, T. T.: Global fire emissions and the contribution of deforestation, savanna, forest, agricultural, and peat fires (1997-2009), Atmospheric Chemistry and Physics, 10, 11707-11735, 10.5194/acp-10-11707-2010, 2010.

van Donkelaar, A., Martin, R. V., Leaitch, W. R., Macdonald, A. M., Walker, T. W., Streets, D. G., Zhang, Q., Dunlea, E. J., Jimenez, J. L., Dibb, J. E., Huey, L. G., Weber, R., and Andreae, M. O.: Analysis of aircraft and satellite measurements from the Intercontinental Chemical Transport Experiment (INTEX-B) to quantify long-range transport of East Asian sulfur to Canada, Atmospheric Chemistry and Physics, 8, 2999-3014, 2008.

Voulgarakis, A., Marlier, M., Faluvegi, G., Shindell, D., Tsigaridis, K., and Mangeon, S.: Interannual variability of tropospheric trace gases and aerosols: The role of biomass burning emissions, Journal of Geophysical Research-Atmospheres, 120, 7157-7173, 10.1002/2014JD022926, 2015.

Wang, Y., Munger, J., Xu, S., McElroy, M., Hao, J., Nielsen, C., and Ma, H.: $CO_2$ and its correlation with CO at a rural site near Beijing: implications for combustion efficiency in China, Atmospheric Chemistry and Physics, 10, 8881-8897, 10.5194/acp-10-8881-2010, 2010.

Warner, J., Carminati, F., Wei, Z., Lahoz, W., and Attie, J.: Tropospheric carbon monoxide variability from AIRS under clear and cloudy conditions, Atmospheric Chemistry and Physics, 13, 12469-12479, 10.5194/acp-13-12469-2013, 2013.

Worden, H. M., Cheng, Y., Pfister, G., Carmichael, G. R., Zhang, Q., Streets, D. G., Deeter, M., Edwards, D. P., Gille, J. C., and Worden, J. R.: Satellite-based estimates of reduced CO and $CO_2$ emissions due to traffic restrictions during the 2008 Beijing Olympics, Geophysical Research Letters, 39, 2012.

Worden, H. M., Deeter, M. N., Frankenberg, C., George, M., Nichitiu, F., Worden, J., Aben, I., Bowman, K. W., Clerbaux, C., Coheur, P. F., de Laat, A. T. J., Detweiler, R., Drummond, J. R., Edwards, D. P., Gille, J. C., Hurtmans, D., Luo, M., Martinez-Alonso, S., Massie, S., Pfister, G., and Warner, J. X.: Decadal record of satellite carbon monoxide observations, Atmospheric Chemistry and Physics, 13, 837-850, 10.5194/acp-13-837-2013, 2013.

Yin, Y., Chevallier, F., Ciais, P., Broquet, G., Fortems-Cheiney, A., Pison, I., and Saunois, M.: Decadal trends in global CO emissions as seen by MOPITT, Atmos. Chem. Phys., 15, 13433-13451, 10.5194/acp-15-13433-2015, 2015.

Yoon, J., Pozzer, A., Hoor, P., Chang, D., Beirle, S., Wagner, T., Schloegl, S., Lelieveld,
J., and Worden, H.: Technical Note: Temporal change in averaging kernels as a source of
uncertainty in trend estimates of carbon monoxide retrieved from MOPITT, Atmospheric
Chemistry and Physics, 13, 11307-11316, 10.5194/acp-13-11307-2013, 2013.
Yoon, J., and Pozzer, A.: Model-simulated trend of surface carbon monoxide for the
2001-2010 decade, Atmospheric Chemistry and Physics, 14, 10465-10482, 10.5194/acp-
551 14-10465-2014, 2014.
Yumimoto, K., Uno, I., and Itahashi, S.: Long-term inverse modeling of Chinese CO
emission from satellite observations, Environmental Pollution, 195, 308-318,
10.1016/j.envpol.2014.07.026, 2014.
Zhang, Q., Streets, D. G., Carmichael, G. R., He, K. B., Huo, H., Kannari, A., Klimont,
Z., Park, I. S., Reddy, S., Fu, J. S., Chen, D., Duan, L., Lei, Y., Wang, L. T., and Yao, Z.
L.: Asian emissions in 2006 for the NASA INTEX-B mission, Atmospheric Chemistry
and Physics, 9, 5131-5153, 2009.
Zhao, Y., Nielsen, C. P., McElroy, M. B., Zhang, L., and Zhang, J.: CO emissions in
China: Uncertainties and implications of improved energy efficiency and emission
control, Atmospheric Environment, 49, 103-113, 10.1016/j.atmosenv.2011.12.015, 2012.


**Table 1:** Description of Simulations

| Simulation | Model | Meteorology | Anthropogenic Emissions | Biomass Burning Emissions |
|---|---|---|---|---|
| G-Ref-C1 | GEOSCCM | internally derived | MACCity | MACCity, GFED3 (2009-2010) |
| C-Ref-C1 | CAM4-Chem | internally derived | MACCity | MACCity, then repeat 2008 |
| Ref-C1-SD | GMI | MERRA | MACCity | Same as GEOSCCM |
| EmFix | GMI | MERRA | Fixed at 2000 | Fixed at 2000 |
| AltEmis | GMI | MERRA | Strode et al [2015] | GFED3 |

**Table 2: Regional Trends and Correlations**

**a. Trends[1,2]**

|  | Years | E. USA | Europe | E. China | N. Hemisphere |
|---|---|---|---|---|---|
| G-Ref-C1[3] | 2000-2010 | -2.2 (0.38) | -1.8 (0.42) | 2.2 (1.1) | -0.76 (3.0) |
| C-Ref-C1[3] | 2000-2010 | -3.4 (0.54) | -2.9 (0.50) | 1.4 (1.4) | -0.90 (3.0) |
| Ref-C1-SD[3] | 2000-2010 | -2.4 (0.53) | -1.6 (0.59) | 1.4 (1.1) | -0.76 (3.0) |
| EmFix[3] | 2000-2010 | 1.3 (0.55) | 1.5 (0.44) | 2.1 (0.87) | 0.96 (2.5) |
| AltEmis[3] | 2000-2010 | 0.71 (0.73) | 0.74 (0.66) | 3.8 (1.4) | 1.1 (3.4) |
| *MOPITT* | *2000-2010* | *-2.5 (0.64)* | *-1.8 (0.69)* | *-2.9 (1.8)* | *-1.4 (2.8)* |
| *MOPITT* | *2000-2014* | *-2.1 (0.41)* | *-1.7 (0.43)* | *-3.1 (1.1)* | *-1.4 (1.7)* |

[1] $10^{16}$ molec cm$^{-2}$ yr$^{-1}$

[2] 1-sigma uncertainty given in parentheses

[3] Simulation results convolved with MOPITT averaging kernel and a priori

**b. Correlation coefficient (r) with monthly MOPITT anomalies[1,2]**

|  | Years | E. USA | Europe | E. China | N. Hemisphere |
|---|---|---|---|---|---|
| G-Ref-C1 | 2000-2010 | **0.26** | **0.39** | 0.061 | **0.71** |
| C-Ref-C1 | 2000-2010 | 0.23 | **0.36** | 0.18 | **0.62** |
| Ref-C1-SD | 2000-2010 | **0.43** | **0.51** | **0.39** | **0.73** |
| EmFix | 2000-2010 | 0.10 | 0.21 | 0.071 | 0.059 |
| AltEmis | 2000-2010 | **0.55** | **0.59** | **0.48** | **0.69** |

[1] Correlations are calculated from the de-trended and de-seasonalized time series.

[2] Statistically significant correlations at the 95% confidence level are indicated in bold.

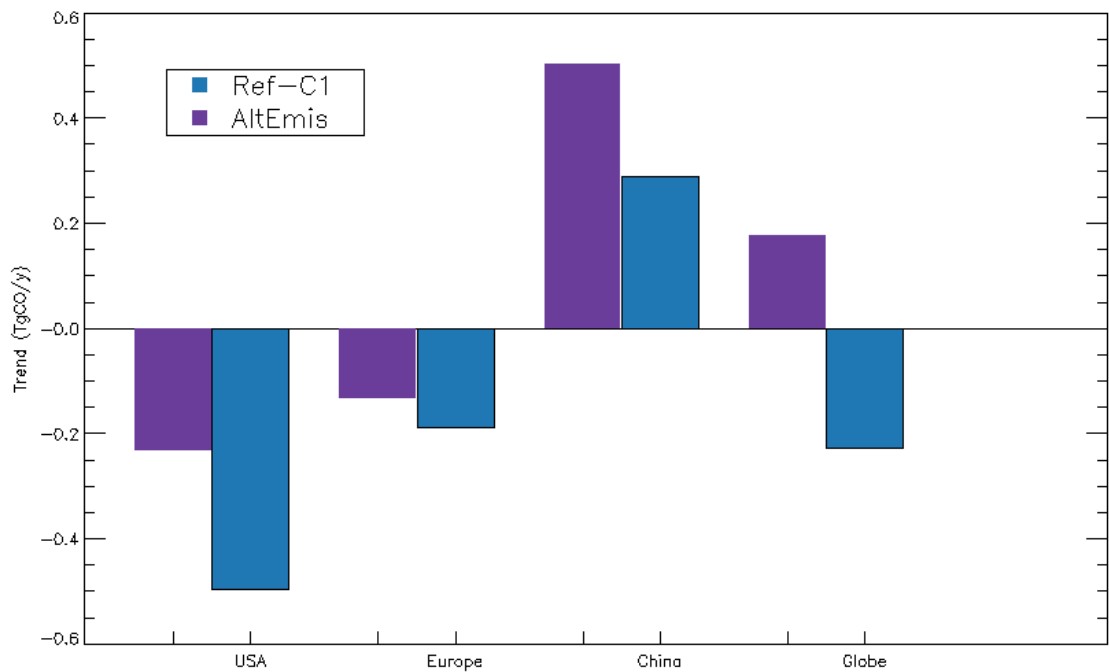

**Figure 1:** Trends in the CO emissions used in the Ref-C1 and Ref-C1-SD simulations (blue bars) and AltEmis simulation (purple bars) over 2000-2010 for the United States, Europe, China, and the world.

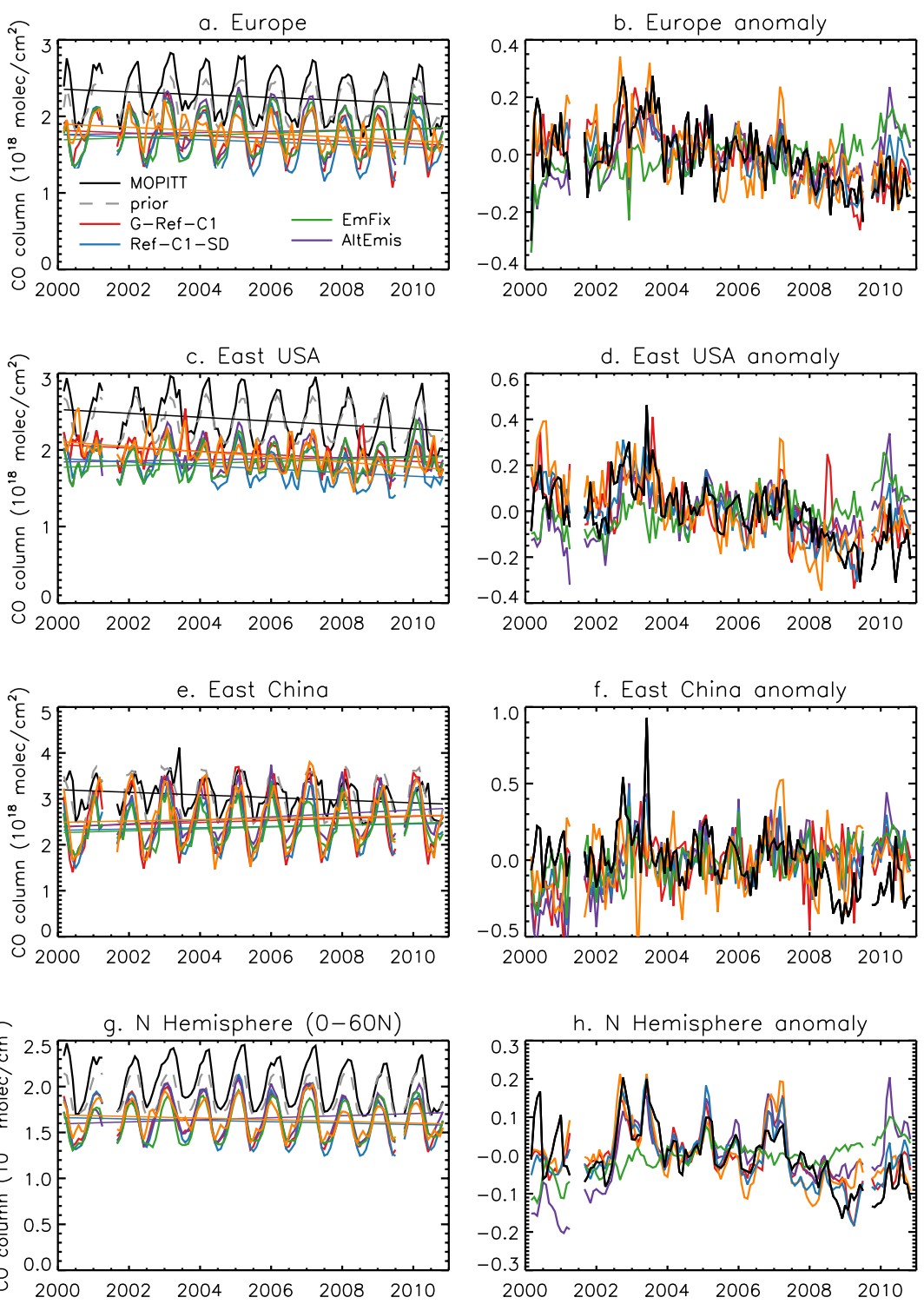

**Figure 2:** The time series and trends (left column) and de-seasonalized monthly anomalies (right column) of the CO column from MOPITT (black), the MOPITT a priori (gray), and simulated by G-Ref-C1 (red), Ref-C1-SD (blue), EmFix (green), C-Ref-C1 (orange), and AltEmis (purple) for 2000-2010. The regions shown are (a,b) Europe (0°-

15°E, 45°-55°N), (c,d) eastern U.S.A. (95°-75°W, 35°-40°N), (e,f) eastern China (110°-123°E, 30°-40°N), and (g,h) the northern hemisphere (0°-60°N).

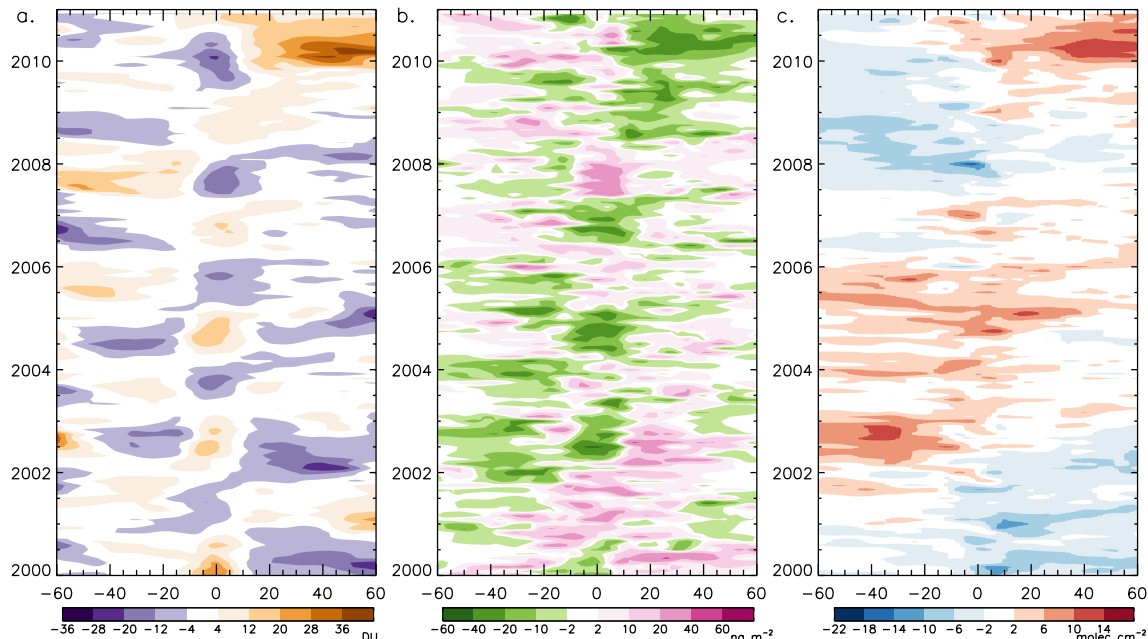

**Figure 3**: Deseasonalized monthly anomalies in the total ozone column (left), mean tropospheric OH (center), and CO column (right) from the EmFix simulation as a function of latitude and month.

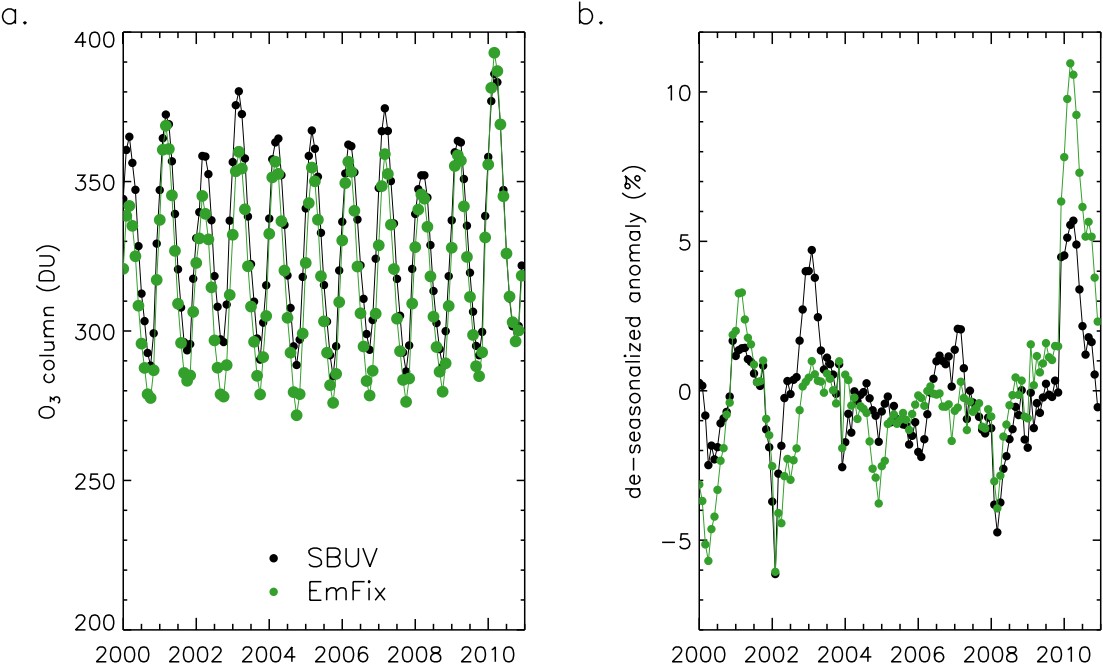

**Figure 4**: Monthly ozone column (a) and de-seasonalized ozone column anomaly (b) in SBUV data (black) and the EmFix simulation (green) for 30°-60°N.

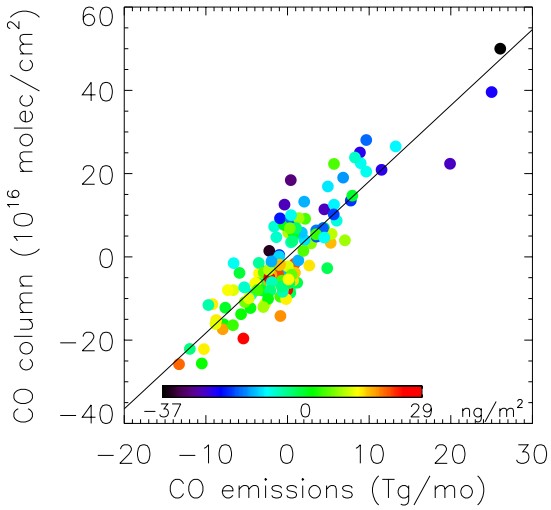

**Figure 5:** Monthly simulated CO column anomalies from the Ref-C1-SD simulation as a function of CO emissions for 10°S-10°N. Colors indicate the simulated OH column anomaly for the given month.