# Peer review of "Interpreting Space-Based Trends in Carbon Monoxide"

_Atmospheric Chemistry and Physics, 2016_

## Referee Comment (RC1) · Anonymous Referee #1 · 30 Mar 2016

The manuscript describes a small series of numerical experiments to understand previously reported trends in column CO from the MOPITT satellite instrument. It is a thoughtful interpretation of the MOPITT data. The manuscript would benefit from more detail in places, detailed below, but it is suitable for publication in ACP.

Detailed comments:

1) The manuscript title would benefit from being more specific.

2) MACCCity or MACCCITY? Be consistent.

3) This reader thought the abstract would benefit from being punchier. What is the punchline? Is it that getting column ozone right is important for understanding column CO trends?

[Figure]

4) Line71. It would be useful for the reader to be more specific about assumptions/data used to build MACCCity. Maybe a few sentences so the reader is not required to immediately chase up details elsewhere.

5) Line section 2.1. This reader believes it would be useful to show an example MOPITT averaging kernel. Just for my own curiosity, is there a difference in the averaging kernel over eastern China during 2010?

6) Line 90: What did the authors assume when calculating the autocorrelation?

7) Line 91: By deseasonalizing the column data by fitting sines/cosines the authors are implicitly assuming stationarity of these data. Do the authors believe this is a valid assumption over a decade-long time series during which the phase and amplitude of the seasonal cycle changes?

8) Line 93: Months with insufficient data? How do the authors define this criterion?

9) Line 95: Do the authors sample the model along the orbit tracks?

10) Minor comment: a few instances where the references should be inline but are not. I expect the typesetting process will pick this up.

11) Section 2.2: Would be useful if the authors provide some regional emission estimates of CO, particularly for pertinent regions.

12) Equation 1. The usual convention is lower-case bold typeface for vectors and upper-case bold typeface for matrices.

13) This reader is confused why the authors included two sets of statistics: 2000-2010 and 2000-2011. If the results from these two sets had been significantly different I would have probably suggested a major re-analysis of the data. But they are not so I suggest (and only suggest) the authors summarize the value of the additional data in a few sentences and report only 2000-2010.

14) Line 186 and elsewhere: The authors won't need reminding that a model-data

correlation r of just over 0.7 is required for the model to describe 50% of the observed variation. In some places the authors extol correlations of X (much less than 0.7) while in other places the author extol the squared correlation values of Y.

15) Line 265: What is responsible for the observed and model total column anomaly in 2010?

16) Line 271: "can be" or "is"?
* * *

---

## Referee Comment (RC2) · Anonymous Referee #2 · 14 Apr 2016

The authors have used a series of chemistry-climate and chemical transport model simulations to understand the negative trends in CO observed by MOPITT. They find that the negative trend in the bottom-up inventories reproduce the trend observed over North America and Europe, but is incapable of capturing the negative trend observed over China. They attributed the discrepancies between the modeled and observed trend in CO over China to changes in the MOPITT vertical sensitivity and to biases in the modeled ozone abundances, which produces a bias in modeled OH and, thus, CO. The paper is well written and the authors did a careful analysis of the trends in the models. I would recommend publication in ACP after the authors have revised the manuscript to address my comments below.

Comments

[Figure]

1) Line 85: The Level 3 MOPITT data are daily or monthly gridded data. Are the authors using the daily gridded data here? Are they using nighttime and daytime, or just daytime data? If they are using Level 3 data, how do they compare the model to the MOPITT data. The model should be sampled at the MOPITT observation locations and times when transforming the model with the MOPITT averaging kernels and a priori profiles. They need to better explain in the paper how this is done.

2) Lines 137 – 141: In giving Equation (1), the authors should explain that the MOPITT retrievals are with respect to the log of the mixing ratio and they should also cite Deeter (2009) "MOPITT (Measurements of Pollution in the Troposphere) Validated Version 4 Product User's Guide" for providing guidance for calculating the column averaging kernels from the averaging kernels that is with respect to the log of the mixing ratio.

3) Lines 201-203: The authors state here that the discrepancy is "driven largely by the failure of the simulations to capture the 2008 dip," but the models are also strongly biased in 2010, for example (Fig. 2f). Indeed, this 2010 bias is the focus of the ozone analysis in Figs. 3 and 4.

4) Please state the regional boundaries for the regions considered in Fig. 2.

5) Lines 226 – 244: I find this discussion confusing. I understand how time-dependent variations in the vertical sensitivity of the MOPITT retrievals could contribute to trends in the data. But I don't understand how, as stated on Lines 228-321, the bias in the modeled CO can produce an artificial trend. It seems to me that there are two possible way this could happen:

a) Are the authors suggesting that changes in the vertical distribution of the model bias, combined with the varying vertical sensitivity of the MOPITT retrievals, produces an artificial trend when the model is convolved with the averaging kernels and a priori profile?

b) If the model bias is constant in time, then the convolved modeled columns should

exhibit the same trend as the data. The presence of a fixed bias in the model together with temporally varying averaging kernels should only impact the trend if the biased model state is so far from the a priori MOPITT state that the linearization assumption in Equation (1) is invalid i.e., the averaging kernels do not accurately capture the sensitivity of the retrieval between the modeled state and the MOPITT a priori. Is this what the authors are trying to say on lines 226-228?

The authors need to explain more clearly how the bias in the model could be contributing to an artificial trend.

6) Lines 261-262: Yes, we expect that anomalies in OH to be inversely related to anomalies in total ozone, however, the OH and ozone anomalies do not seen to be strongly correlated in Fig. 3. It would be helpful if the authors gave the correlation coefficient between the two quantities for different latitude bands in the tropics and extratropical northern hemisphere.

7) Lines 263-264: The ozone column anomaly in Fig. 3 is in the extratropical northern hemisphere, mainly in early (Jan-Mar) 2010. Although the global, annual mean CO lifetime is 1-2 months, in the extratropics in winter it could be longer than a season. If that is the case, it is unclear to me how the changes in OH in early 2010 could drive such large changes in CO between 30-60N in winter.

8) Figure 4: The 2010 ozone anomaly is about 5%. What are the altitude ranges that are contributing to this bias in the column? Are these changes mainly in the UTLS?

Technical comments

1. Line 64: Change "results of (Li and Liu, 2010)" to "results of Li and Liu (2010"

2. Figure 2: It difficult to see the seven different lines in each panel. If the authors remove the titles on the y-axes on the panels in the right column and reduce the spacing between the panels, it may be possible to enlarge each panel to make the plots more legible.

3. Figure 3: What are the units for the colorbars?

---

## Author Comment (AC1) · 16 May 2016

**Response to Referee Comments on "Interpreting Space-Based Trends in Carbon Monoxide"**

Reviewer comments are in blue, and the response in black.

*Response to Reviewer 1:*

The manuscript describes a small series of numerical experiments to understand pre- viously reported trends in column CO from the MOPITT satellite instrument. It is a thoughtful interpretation of the MOPITT data. The manuscript would benefit from more detail in places, detailed below, but it is suitable for publication in ACP.

We thank the referee for the review and respond to individual comments below.

Detailed comments:

1) The manuscript title would benefit from being more specific.

We updated the title to: "Interpreting Space-Based Trends in Carbon Monoxide with Multiple Models"

2) MACCCity or MACCCITY? Be consistent.

We now use MACCity throughout the manuscript.

3) This reader thought the abstract would benefit from being punchier. What is the punchline? Is it that getting column ozone right is important for understanding column CO trends?

We added the following sentences to the abstract to emphasize our conclusions: "This demonstrates that biases in a model's average concentrations can influence the interpretation of the temporal trend compared to satellite observations." and "These results demonstrate that accurately simulating variability in the ozone column is important for simulating and interpreting trends in CO."

4) Line71. It would be useful for the reader to be more specific about assumptions/data used to build MACCCity. Maybe a few sentences so the reader is not required to immediately chase up details elsewhere.

We added the following details in section 2.2: "The MACCity inventory linearly interpolates the decadal anthropogenic emissions from the ACCMIP inventory (Lamarque et al., 2010) for 2000, and the RCP8.5 emissions for 2005 and 2010, to each year in between.  The MACCity biomass burning emissions have year-to-year variability based on the GFED-v2 (van der Werf et al., 2006) inventory."

5) Line section 2.1. This reader believes it would be useful to show an example MO-PITT averaging kernel. Just for my own curiosity, is there a difference in the

We added a supplemental figure showing the column averaging kernels for 2010 and two other years.  We added this discussion to Section 2.1: "Supplemental Figure S1 shows the MOPITT column averaging kernels averaged over four regions.  The column averaging kernels depend on the observed scene, and vary year to year as well as seasonally.  The dependence of the column averaging kernels on the CO mixing ratio profile (Deeter, 2009) explains the high values in the lower troposphere over eastern China in winter."   Although the kernel over eastern China varies year to year, we did not find 2010 to be fundamentally different from other years.

6) Line 90: What did the authors assume when calculating the autocorrelation?

We clarify that we calculated the autocorrelation for a 1-month lag.

7) Line 91: By deseasonalizing the column data by fitting sines/cosines the authors are implicitly assuming stationarity of these data. Do the authors believe this is a valid assumption over a decade-long time series during which the phase and amplitude of the seasonal cycle changes?

Although there are anomalies in particular months that cause year-to-year variability in the seasonal cycle, we did not find a systematic change in the phase and amplitude of the seasonal cycle.

8) Line 93: Months with insufficient data? How do the authors define this criterion?

We now clarify: "Months with no MOPITT data or only a few days of MOPITT data are excluded from the trend analysis.  This includes May-August of 2001 and August-September of 2009."

9) Line 95: Do the authors sample the model along the orbit tracks?

We use the level 3 data, which is a gridded product.  We added this information in section 2.1.

10) Minor comment: a few instances where the references should be inline but are not. I expect the typesetting process will pick this up.

Fixed.

11) Section 2.2: Would be useful if the authors provide some regional emission estimates of CO, particularly for pertinent regions.

We added the following information: "From 2000 to 2010, CO emissions in the MACCity inventory decreased from 31 to 11 Tg $yr^{-1}$ over the eastern U.S., from 97 to 59 Tg $yr^{-1}$ over Europe, and increased from 56 Tg to 72 Tg $yr^{-1}$ over eastern China."

12) Equation 1. The usual convention is lower-case bold typeface for vectors and upper-case bold typeface for matrices.

We modified the equation to reflect this convention.

13) This reader is confused why the authors included two sets of statistics: 2000-2010 and 2000-2011. If the results from these two sets had been significantly different I would have probably suggested a major re-analysis of the data. But they are not so I suggest (and only suggest) the authors summarize the value of the additional data in a few sentences and report only 2000-2010.

We updated the text and figures to report only values for 2000-2010, and removed the supplemental figure, which showed results for 2000-2011.  We added the following sentence to Section 2.2: "Some simulations were available through 2011, while others ended in 2010.  We therefore report results for 2000-2010, but note that extending the analysis through 2011 does not alter the conclusions."

14) Line 186 and elsewhere: The authors won't need reminding that a model-data correlation r of just over 0.7 is required for the model to describe 50% of the observed variation. In some places the authors extol correlations of X (much less than 0.7) while in other places the author extol the squared correlation values of Y.

We agree that many of the r values shown in Table 2b are below 0.7 and thus imply that the simulations are capturing less than half of the variance.  However, we find it useful to report and discuss these r values to indicate the relative performance of different simulations.  We updated this table to indicate which correlations are statistically significant.  We updated the text in Section 3.3 to report r values rather than $r^2$ for consistency with the rest of the paper.

15) Line 265: What is responsible for the observed and model total column anomaly in 2010?

Steinbrecht et al. (2011) attribute the 2010 anomaly in northern midlatitude ozone observations to a combination of an unusually strong negative Arctic Oscillation and North Atlantic Oscillation and the easterly phase of the quasi-biennial oscillation.  We added this information in Section 3.3.

16) Line 271: "can be" or "is"?

We changed this to "is partially".

*Response to Referee 2:*

The authors have used a series of chemistry-climate and chemical transport model simulations to understand the negative trends in CO observed by MOPITT. They find that the negative trend in the bottom-up inventories reproduce the trend observed

over North America and Europe, but is incapable of capturing the negative trend observed over China. They attributed the discrepancies between the modeled and observed trend in CO over China to changes in the MOPITT vertical sensitivity and to biases in the modeled ozone abundances, which produces a bias in modeled OH and, thus, CO. The paper is well written and the authors did a careful analysis of the trends in the models. I would recommend publication in ACP after the authors have revised the manuscript to address my comments below.

We appreciate the thoughtful review and respond to comments below.

Comments

1) Line 85: The Level 3 MOPITT data are daily or monthly gridded data. Are the authors using the daily gridded data here? Are they using nighttime and daytime, or just daytime data? If they are using Level 3 data, how do they compare the model to the MOPITT data. The model should be sampled at the MOPITT observation locations and times when transforming the model with the MOPITT averaging kernels and a priori profiles. They need to better explain in the paper how this is done.

We now clarify in section 2.1 that we are using the monthly gridded daytime data, and that the level 3 product includes the averaging kernel and a priori for each grid box.  In section 2.2, we added that we regrid the model results to the MOPITT grid. We expect the error from using monthly mean simulated CO instead of sampling at the overpass time to be small since CO does not have a large diurnal cycle.  Our analysis includes a free-running CCM simulation as well as CTM simulations.  The meteorology of the free-running CCM will not match up with the observed for individual days, and our focus is on monthly and interannual rather than daily variability, so we chose to use the monthly mean MOPITT product.  Martinez-Alonso et al [2014] demonstrated that there is only a small bias from using gridded average averaging kernels rather than the kernels of individual retrievals.

2) Lines 137 – 141: In giving Equation (1), the authors should explain that the MOPITT retrievals are with respect to the log of the mixing ratio and they should also cite Deeter (2009) "MOPITT (Measurements of Pollution in the Troposphere) Validated Version 4 Product User's Guide" for providing guidance for calculating the column averaging kernels from the averaging kernels that is with respect to the log of the mixing ratio.

We added the following line to Section 2.2: "The column averaging kernel is calculated from the standard averaging kernel matrix, which is based on the log of the CO concentration profile, following the method of Deeter (2009):

$$a_j = (K / \log_{10} e) \sum \Delta p_i \, v_{rtv,i} \, A_{ij} \qquad (2)$$

where $\Delta p_i$ and $v_{rtv,i}$ are the pressure thickness and retrieved CO concentration, respectively, of level i, $\mathbf{A}$ is the standard averaging kernel matrix, and $K = 2.12 * 10^{13}$

molec cm$^{-2}$ hPa$^{-1}$ ppb$^{-1}$."

3) Lines 201-203: The authors state here that the discrepancy is "driven largely by the failure of the simulations to capture the 2008 dip," but the models are also strongly biased in 2010, for example (Fig. 2f). Indeed, this 2010 bias is the focus of the ozone analysis in Figs. 3 and 4.

We added: "leading to an overestimate that continues through 2010".

4) Please state the regional boundaries for the regions considered in Fig. 2.

We added this information to the caption of Fig. 2.

5) Lines 226 – 244: I find this discussion confusing. I understand how time-dependent variations in the vertical sensitivity of the MOPITT retrievals could contribute to trends in the data. But I don't understand how, as stated on Lines 228-321, the bias in the modeled CO can produce an artificial trend. It seems to me that there are two possible way this could happen:

a) Are the authors suggesting that changes in the vertical distribution of the model bias, combined with the varying vertical sensitivity of the MOPITT retrievals, produces an artificial trend when the model is convolved with the averaging kernels and a priori profile?

b) If the model bias is constant in time, then the convolved modeled columns should exhibit the same trend as the data. The presence of a fixed bias in the model together with temporally varying averaging kernels should only impact the trend if the biased model state is so far from the a priori MOPITT state that the linearization assumption in Equation (1) is invalid i.e., the averaging kernels do not accurately capture the sen- sitivity of the retrieval between the modeled state and the MOPITT a priori. Is this what the authors are trying to say on lines 226-228?

The authors need to explain more clearly how the bias in the model could be contribut- ing to an artificial trend.

As the reviewer suggests in a), the bias in CO varies with altitude, so if the vertical sensitivity described by the averaging kernel changes to e.g. place more weight on higher altitudes, this will change the value of the convolved CO column even if there were no changes in the CO profile.  But even if the bias were constant with altitude, changes in the averaging kernel result in more or less weight placed on the a priori versus the CO simulated by the model.  Thus, a difference between the a priori and the model means that placing more (or less) weight on the a priori will change the resulting value of $C_{sim}$.  Since the a priori profiles and columns are constant in time, taking the time derivative of equation 1 yields:

$$\partial C_{sim}/\partial t = \mathbf{a}\,(\partial \mathbf{x}_{mod}/\partial t) + \partial \mathbf{a}/\partial t\,(\mathbf{x}_{mod} - \mathbf{x}_0)$$

The second term on the right hand side shows that the larger the bias between the modeled CO and the a priori, the larger the impact of the changing averaging kernel. We now discuss this in Section 3.2.

6) Lines 261-262: Yes, we expect that anomalies in OH to be inversely related to anomalies in total ozone, however, the OH and ozone anomalies do not seen to be strongly correlated in Fig. 3. It would be helpful if the authors gave the correlation coefficient between the two quantities for different latitude bands in the tropics and extratropical northern hemisphere.

We added the following sentence to this discussion: "The correlation coefficient between OH and column ozone is -0.53 for the 15°S-15°N average, -0.72 for the 15°-25°N average, and -0.75 for the 30°-60°N average."

7) Lines 263-264: The ozone column anomaly in Fig. 3 is in the extratropical northern hemisphere, mainly in early (Jan-Mar) 2010. Although the global, annual mean CO lifetime is 1-2 months, in the extratropics in winter it could be longer than a season. If that is the case, it is unclear to me how the changes in OH in early 2010 could drive such large changes in CO between 30-60N in winter.

The early 2010 OH anomaly occurs in the northern tropics as well as the extratropics. We added the following sentence to highlight this: "This OH anomaly extends from the northern tropics to the midlatitudes." Northern midlatitude (30-60N) CO and OH are anticorrelated with an r value of -0.69. This r value increases to -.78 if we apply a three-month smoothing, reflecting the longer lifetime of CO. However, since the lifetime of CO is several months, we do not expect a one-to-one correspondence between CO anomalies and OH anomalies, and we added a sentence stating this. We also clarified figure 3 by adding the units to each panel.

8) Figure 4: The 2010 ozone anomaly is about 5%. What are the altitude ranges that are contributing to this bias in the column? Are these changes mainly in the UTLS?

The 2010 anomaly in both the SBUV data and the model is driven by ozone at pressures higher than 25hPa. The vertical resolution of the SBUV data does not allow us to determine the specific altitude of the bias. However, comparison to MLS data shows that GMI has a high bias in lower stratospheric ozone (pressures greater than 50 hPa) in the first half of 2010.

Technical comments

1. Line 64: Change "results of (Li and Liu, 2010)" to "results of Li and Liu (2010)"

Fixed

2. Figure 2: It difficult to see the seven different lines in each panel. If the authors remove the titles on the y-axes on the panels in the right column and reduce the spacing between the panels, it may be possible to enlarge each panel to make the

plots more legible.

We updated this figure as suggested.

---

## Author Response (AR2)

**Response to Editor**

The comments from the editor are in blue, and our response is in black.

Comments to the Author:
The authors have done a good job of responding to the reviewers' concerns. The authors have done a nice job of looking at the different factors which cause anomalous CO trends in models relative to observations. This should be acceptable for publication in ACP subject to minor revisions:

We appreciate the thoughtful feedback from the editor. We respond to the specific comments below, which have helped strengthen our conclusions.

You make the statement that the changing balance of local v. hemispheric CO is insufficient to explain the trends in CO. I wonder if it is possible to quantify the contributions from each of these sources to the anomalous trends? At least to approximate (a) the hemispheric contribution,

Comparison of the trend for the northern hemisphere to that of eastern China can provide an estimate of the hemispheric contribution. We added the following sentence to the Conclusions: "Indeed, the negative trend in MOPITT CO over eastern China ($-2.9*10^{16}$ molec cm$^{-2}$ yr$^{-1}$) is stronger than that of the northern hemisphere average ($-1.4*10^{16}$ molec cm$^{-2}$ yr$^{-1}$), indicating that changes in hemispheric CO account for less than half of the trend over China."

(b) how the mean bias applied to the averaging kernel (equation 3), (c) anomalies in Ozone, and (d) emissions might be contributing.

We added the following statement to Section 3.2 to quantify the effect of the mean bias applied to the averaging kernel: "Comparing the trend for the constant averaging kernel case with the original simulated trend for Ref-C1-SD ($1.4*10^{16}$ molec cm$^{-2}$ yr$^{-1}$) suggests that the changing averaging kernels combined with the model bias contribute $0.84*10^{16}$ molec cm$^{-2}$ yr$^{-1}$ to the simulated trend." To better quantify the role of chemistry (including ozone anomalies) and transport versus emissions, we now state "Subtracting the EmFix trend from the Ref-C1-SD trend shows that the changing emissions contribute a CO trend of -0.7 molec cm$^2$ yr$^{-1}$ over eastern China. The 2.1 molec cm$^2$ yr$^{-1}$ trend in the EmFix simulation, which reflects the impacts of the simulated chemistry and transport, thus contributes to the erroneous sign of the trend in the GMI simulations." However, our simulations do not allow us to separate the role of ozone anomalies from all other chemical and transport effects.

This would strengthen the conclusions if it is possible to pull this out of your analysis. It could just be stated somewhere in a few sentences if possible.

[revised manuscript text omitted]